# Survival in Breast Cancer Patients with Bone Metastasis: A Multicenter Real-World Study on the Prognostic Impact of Intensive Postoperative Bone Scan after Initial Diagnosis of Breast Cancer (CSBrS-023)

**DOI:** 10.3390/cancers14235835

**Published:** 2022-11-26

**Authors:** Liu Yang, Wei Du, Taobo Hu, Miao Liu, Li Cai, Qiang Liu, Zhigang Yu, Guangyu Liu, Shu Wang

**Affiliations:** 1Department of Breast Disease Center, Peking University People’s Hospital, Beijing 100044, China; 2Department of Breast Oncology, Harbin Medical University Cancer Hospital, Harbin 150000, China; 3Department of Breast Surgery, Sun Yai-Sen Memorial Hospital, Guangzhou 510120, China; 4Department of Breast Surgery, The Second Hospital of Shandong University, Jinan 250021, China; 5Department of Breast Surgery, Fudan University Shanghai Cancer Center, Shanghai 200032, China

**Keywords:** breast cancer, bone metastases, bone scan, follow-up, prognosis, survival

## Abstract

**Simple Summary:**

The bone scan (BS) is widely used in follow-up to detect bone metastasis (BM) in breast cancer (BC) patients presenting bone-related symptoms after surgery. However, it remains controversial whether asymptomatic BS (intensive postoperative BS) screening could be translated into a survival benefit. Therefore, we conducted this multicenter real-world study to understand the prognostic impact of intensive postoperative BS screening among 1059 Chinese patients with BM during the years 2005–2013. This study showed that intensive postoperative BS screening was an independent prognostic factor and prolonged the survival in patients with BC with BM. The prognostic value of intensive BS screening was consistently favorable for survival in patients at clinical high-risk. These findings suggested that intensive BS screening was important for improving survival, and should be recommended for postoperative surveillance, especially for patients with a high risk of recurrence and metastasis.

**Abstract:**

The prognostic value of intensive postoperative bone scan (BS) screening, which is performed in asymptomatic patients with breast cancer (BC) after surgery, remained unclear. Patients diagnosed with BC with bone metastasis (BM) from five medical centers in China during the years 2005–2013 were retrospectively collected. Propensity score matching (PSM) was performed to balance the baseline characteristics. The survival outcomes were overall survival (OS) and overall survival after BM (OSABM). Among 1059 eligible patients, 304 underwent intensive postoperative BS while 755 did not. During a median follow-up of 6.67 years (95%CI 6.45, 7.21), intensive postoperative BS prolonged the median OS by 1.63 years (Log-Rank *p* = 0.006) and OSABM by 0.66 years (Log-Rank *p* = 0.002). Intensive postoperative BS was an independent prognostic factor for both OS (adjusted HR 0.77, 95%CI 0.64, 0.93, adjusted *p* = 0.006) and OSABM (adjusted HR 0.71, 95%CI 0.60, 0.86, adjusted *p* < 0.001). The prognostic value of intensive postoperative BS was consistently favorable for OS among clinical high-risk patients, including those with ages younger than 50, stage II, histology grade G3 and ER-Her2- subtype. This multicenter real-world study showed that intensive postoperative BS screening improved survival for BC patients with BM and should probably be recommended for postoperative surveillance, especially for patients at clinical high-risk.

## 1. Introduction

Breast cancer (BC) is the most commonly diagnosed malignant cancer in women [1], and bone is the most common distant metastatic site [2,3,4]. The bone scan (BS), a conventional and cost-effective modality for detecting the entire skeleton in one examination [5,6], is widely used in postoperative follow-up for surveillance of bone metastasis (BM) in BC patients presenting related symptoms after surgery. However, current guidelines do not recommend intensive BS screening, which is referred to BS screening in asymptomatic patients, without specific findings on clinical examination before a diagnosis of BM.

The prognostic value of intensive postoperative BS remains unclear. Two well-designed randomized controlled trials, GIVIO (Interdisciplinary Group for Cancer Care Evaluation) trials [7], as well as Rosselli del Turco trials [8], and the Cochrane meta-analysis [9] demonstrated that intensive follow-up (imaging examinations including BS and laboratory tests) does not improve overall survival compared to clinical follow-up (physical examinations and annual mammography). Hence the American Society of Clinical Oncology (ASCO) [10], National Comprehensive Cancer Network (NCCN) [11] and European Society for Medical Oncology (ESMO) [12] do not recommend an intensive follow-up including BS.

It is important to note that the two trials were conducted almost three decades ago when advanced postoperative screening methods and palliative therapeutic options were scarce. Moreover, oncologists at that time lacked an adequate understanding of the intrinsic biological characteristics of BC. Recently, new regimens of systemic chemotherapy [13,14] and endocrine therapy [15] have made considerable progress in increasing patients’ survival with far-advanced cancer. Anti-Her2 (human epidermal growth factor receptor 2) therapy increased the prognosis of patients with Her2-positive metastatic BC [16,17]. Bone-modifying agents, such as bisphosphonates [18] and denosumab [19], slowed down the progression of skeletal-related events, thus promoting the quality of life.

It is possible that recent improvements in diagnostics and treatments could promote earlier detection and effective treatment of BM, important for improving survival. Therefore, we conducted this multicenter real-world study to understand the prognostic factors of BC patients with BM, especially the prognostic impact of an intensive postoperative BS after initial diagnosis of BC.

## 2. Materials and Methods

### 2.1. Design and Patients

According to Chinese Society of Breast Surgery (CSBrS), this multicenter real-world study was conducted by five medical centers in China. This study has been registered in Clinicaltrials.gov as NCT03924609 on 23 April 2019 and approved by the Ethics Committee of the People’s Hospital of Peking University (No. 2021PHB071-001). As this study was a retrospective study and all data were performed anonymously, the need for informed consent from patients was waived. All data generated or analyzed during the study are included in the published paper.

Patients eligible were required to have a histology-confirmed diagnosis of invasive BC and undergo curative-intent primary therapy. The diagnosis of BM must be supported by pathological or imaging evidence. The following cases were not eligible: (1) with other malignant primary cancer; (2) de novo stage IV BC; (3) incomplete and ambiguous clinical and pathological records.

### 2.2. Clinicopathological Factors

Clinicopathological factors of eligible patients were extracted from the standardized case report forms. Intensive postoperative BS was defined as at least one asymptomatic postoperative BS screening after initial diagnosis of BC before a diagnosis of BM. Clinical postoperative BS was referred to postoperative BS screening only performed in patients presenting bone-related symptoms. Primary tumor staging was defined according to the criteria of the TNM (tumor-nodal-metastasis) staging system by AJCC (American Joint Committee on Cancer) [20]. The histology type of BC was defined according to criteria from the WHO (World Health Organization) [21]. The molecular subtypes of BC were classified based on the expression of the estrogen receptor (ER) and Her2 according to ASCO/CAP (American Society of Clinical Oncology/College of American Pathologists) [22,23]. Based on the timing of BM and visceral metastasis (VM), the pattern of distant metastasis was mainly divided into the following types: (1) BM only: only diagnosed with BM; (2) BM with VM: diagnosed with BM and VM simultaneously; (3) BM to VM: first diagnosed with BM, followed by VM; (4) VM to BM: first diagnosed with VM, followed by BM.

### 2.3. Follow-Up and Outcomes Definition

Follow-up was conducted by telephone or clinical visit from the date of diagnosis of BM until death. The follow-up information was obtained from the databases of the participating medical centers. The survival endpoints were overall survival (OS), which was calculated from the date BC was diagnosed to the date of death, and overall survival after diagnosis of bone metastasis (OSABM), which was calculated from the date BM was diagnosed to the date of death. The length of bone-metastasis free interval (BMFI) was also retrospectively observed, which was calculated as the time from diagnosis of BC to initial BM.

### 2.4. Propensity Score Matching (PSM)

When comparing survival between patients who underwent an intensive postoperative BS and those who underwent a clinical postoperative BS, propensity score matching was used to balance the baseline characteristics. We performed a 1:2 nearest-neighbor matching procedure within a caliper of 0.02 and all clinic and pathological factors were included in the matching model. Balance between the two groups before and after matching was assessed using standardized mean differences (SMD) and *p*-value by chi-square test or *t* test. SMD > 0.20 or *p*-value < 0.05 were considered imbalanced.

### 2.5. Statistical Analysis

Continuous variables were reported as mean and standard deviation, whereas categorical variables were reported as percentage. Statistical differences in the distribution of continuous and categorical variables were conducted by *t*-test and chi-square test, respectively. The statistical differences in the distribution of BMFI in various subgroups according to TNM stage and molecular subtype of BC were tested using the Kruskal–Wallis method.

Survival analysis was performed using the Kaplan–Meier method before and after PSM, thus median survival time was estimated and the Log-rank test was used for comparisons between groups. After PSM, univariate and multivariate Cox proportional hazards regression analyses and associated 95% confidence intervals (95%CI) were used to assess whether the hazard risks of survival endpoints in patients varied by certain clinical or pathological factors. Factors that showed a univariate connection with survival (*p*-value < 0.20) or considered clinically relevant were entered into the multivariate Cox proportional hazard regression model. Interaction terms were tested using the qualitative method and the univariate stratified Cox proportional hazard regression model, which were used to investigate whether the association between postoperative follow-up strategies and survival outcomes differed according to all clinical and pathological factors. Two-tailed *p*-values < 0.05 were considered statistically significant. All analyses were conducted using R*64 4.0.0 (Beijing, China, http://Rproject.org, accessed on 10 January 2022) and IBM SPSS Statistics 25.

## 3. Results

### 3.1. Patient Characteristics

From February 2005 to December 2013, we retrospectively identified 1425 patients with BC with BM from five medical centers in China. Excluding 239 patients with de novo stage IV BC and 127 with incomplete clinicopathological records, 1059 eligible patients were included in the analyses. The flow chart of the process of patients’ enrollment and analyses is presented in Figure 1.

Among 1059 eligible patients, 304 underwent an intensive postoperative BS while 755 underwent a clinical postoperative BS. The median time when a patient received the first intensive postoperative BS was 2.5 years after initial diagnosis of BC. Baseline characteristics in the two groups stratified by postoperative follow-up strategy were balanced after PSM (shown in Table 1).

### 3.2. The Impact of an Intensive Postoperative BS on Survival

Follow-up was regularly performed until December 2018. During a median follow-up of 6.67 years (95%CI 6.45, 7.21), 759 out of 1059 eligible patients were dead: 197 in the intensive postoperative BS group and 562 in the clinical postoperative BS group. Before PSM, both median OS and OSABM of patients with an intensive postoperative BS were longer than those with a clinical postoperative BS (median OS, 7.99 vs. 6.61 years, Log-Rank *p* = 0.003, Figure 2A; median OSABM, 3.16 vs. 2.57 years, Log-Rank *p* = 0.003, Figure 2C). After PSM, both OS and OSABM benefits were still statistically significant in patients with an intensive postoperative BS (median OS, 7.88 vs. 6.25 years, Log-Rank *p* = 0.006, Figure 2B; median OSABM, 3.16 vs. 2.50 years, Log-Rank *p* = 0.002, Figure 2D).

### 3.3. Univariate and Multivariate Analysis of Factors Influencing Survival

When adjusting clinicopathological covariates after PSM, intensive postoperative BS was a favorable prognostic factor for both OS and OSABM of patients with BC with BM and reduced the risk of mortality by 23% (OS, adjusted HR 0.77, 95%CI 0.64, 0.93, adjusted *p* = 0.006; OSABM, adjusted HR 0.71, 95%CI 0.60, 0.86, adjusted *p* < 0.001). Histology type, TNM stage, distant metastatic pattern and palliative endocrine therapy were also independent prognostic factors for both OS and OSABM. Additionally, BMFI and age at diagnosis of BM were independent prognostic factors of OS and OSABM, respectively. The results of univariate and multivariate analysis of clinicopathological factors affecting OS and OSABM among eligible patients after PSM are listed in Table 2.

### 3.4. Interaction and Univariate Stratified Analysis of the Impact of an Intensive Postoperative BS on Survival

As shown in Figure 3, eligible patients were stratified by all clinicopathological factors and palliative treatment methods on BM to explore the relationship between postoperative follow-up strategy and survival after PSM. The prognostic value of an intensive postoperative BS was consistently favorable for OS among BC patients at clinical high-risk, including an age at diagnosis of BM younger than 50, TNM stage II, histology grade G3 and ER-Her2-subtype (Figure 3A). Similarly, as for OSABM, the favorable prognostic value of an intensive postoperative BS was also significant in patients at clinical high-risk, including TNM stage II, histology grade G3 and ER-Her2-subtype (Figure 3B).

### 3.5. The Impact of Palliative Treatments on Survival Stratified by Molecular Subtype

From the point of molecular subtypes of BC, we observed the association between palliative treatments and survival of patients with BM. For patients with a Her2+ BC, 50% (94/188) received palliative anti-Her2 therapy. Palliative anti-Her2 therapy prolonged median OS by 2.4 years (Log-Rank *p* = 0.002; HR 0.60, 95%CI 0.43, 0.83; Figure 4A) and OSABM by 1.6 years (Log-Rank *p* < 0.001; HR 0.49, 95%CI 0.35, 0.68; Figure 4B) among Her2+ patients. For patients with an ER + BC, 75.5% (545/722) underwent palliative endocrine therapy. Palliative endocrine therapy improved both OS (Log-Rank *p* < 0.001; HR 0.70, 95%CI 0.57, 0.86; Figure 4C) and OSABM (Log-Rank *p* = 0.007; HR 0.75, 95%CI 0.61, 0.92; Figure 4D) for this subgroup of patients. In addition, 87.7% (222/253) of patients with an ER-HER2-BC received palliative chemotherapy. However, palliative chemotherapy converted into neither OS (Log-Rank *p* = 0.300; HR 0.81, 95%CI 0.53, 1.24; Figure 4E) nor OSABM (Log-Rank *p* = 0.070; HR 0.68, 95%CI 0.45, 1.04; Figure 4F) benefits for ER-Her2-patients.

### 3.6. The Association of BMFI with BC Stage and Molecular Subtype

The median BMFI was 3.08 years for 1059 eligible patients. However, as shown in Figure 5, BC patients with a different TNM stage and molecular subtype presented specific distributions of the length of BMFI. The median BMFI was 3.29 years for patients at stage I-II and 2.13 years for patients at stage III (*p* < 0.001, Figure 5C). The annual incidence of BM reached a peak at the second year after initial diagnosis of BC among patients at stage III (24.5%, 120/489), the third year among patients at stage II (19.0%, 84/443), while the fourth year among patients at stage I (18.1%, 23/127, Figure 5A). The median BMFI was 3.38, 2.88 and 2.30 years for patients with an ER+, ER-Her2- and Her2+ BC, respectively (*p* < 0.001, Figure 5D). Compared with ER+ and ER-Her2-, patients with a Her2+ BC progressed to BM more rapidly. The cumulative incidence of BM (two years after initial diagnosis of BC) was 26.6% (192/722) for ER+ patients and 34.4% (87/253) for ER-Her2-patients; however, it was 42.0% (79/188) for Her2+ patients (Figure 5B).

## 4. Discussion

This multicenter real-world study showed an intensive postoperative BS improved survival for BC patients with BM. In the point of molecular subtypes of BC, palliative anti-Her2 therapy and endocrine therapy improved both OS and OSABM among patients with a Her2+ and ER+ BC, respectively. These results indicated that the intensive postoperative BS and phenotype-specific palliative systemic treatments were important for improving survival of patients with BM.

Currently, ASCO, NCCN and ESMO guidelines do not recommend an intensive postoperative BS for BC patients [10,11,12]. However, in clinical practice, there are substantial variations in adherence to guideline recommendations. Intensive follow-up is a widespread reality that costs 2.2–3.6 times more than follow-up suggested by guidelines [24]. In a large population-based retrospective longitudinal study (*n* = 11,219) of women in Canada, 8.7–14.6% of women underwent BS screening in each follow-up year, and about half of them had greater than ASCO guideline-recommended surveillance imaging for metastatic diseases [25]. In line with these results, Surveillance, Epidemiology, and End Results (SEER)-Medicare database showed that 13.3% of 37,967 patients underwent at least one BS screening in the first year of follow-up [26]. Similarly, in our study, 28.7% (304/1059) of patients received an intensive postoperative BS. There are several possible reasons for the overuse of intensive BS imaging. First, the patient-driven anxiety and the feeling of reassurance induced by intensive postoperative surveillance, including the BS. Stemmler et al. have examined 801 questionnaires of German women with a history of BC and reported that more than 47.8% of them needed an intensive schedule, which increased their feeling of security [27]. Second, patients with early or limited metastatic recurrence may be curable; thus, the monitoring of asymptomatic patients could result in better efficacy of BC treatment, at least in theory, when tumor burden is low [26]. Third, all the high-level evidence was conducted almost 30 years ago in an era of outdated technology and limited therapeutic options. Current evidence demonstrated that improvements in diagnostics and treatments could improve the survival of patients with metastatic BC, especially with more detailed subtype classification and corresponding efficient target therapies [13,14,15,17]. However, there are no current well-designed trials to verify this issue. To the best of our knowledge, this is the first study that observed the prognostic value of an intensive postoperative BS in patients with BC with BM.

In our study, an intensive postoperative BS resulted in an independent prognostic factor of OS and OSABM among patients with BC with BM. It was worth noticing that 85.4% (904/1059) of patients received palliative chemotherapy, and 66.1% (700/1059) received bone-modifying therapy. In addition, 75.5% (545/722) of ER+ patients received palliative endocrine therapy and 50% (94/188) of Her2+ patients received palliative anti-Her2 therapy. The strength of these treatments was much stronger than it was decades ago. Palliative endocrine therapy had been identified as an independent prognostic factor for OS as well as OSABM, and palliative anti-Her2 therapy also improved OS and OSABM of patients with Her2+ BC. For ER-Her2-patients, palliative systemic chemotherapy increased 5-year OS by 14.3% (57.7% vs. 43.4%) and 2-year OSABM by 18.7% (49.7% vs. 31.0%) compared with the patients who did not receive palliative chemotherapy. This evidence suggested that intensive detection and effective phenotype-specific systemic intervention for BM could be translated into a survival benefit.

In order to make intensive postoperative BSs more cost-effective, we selected high-risk patients based on stratified analysis. A higher tumor burden led to a higher risk of distant metastasis [28,29,30,31]. Our study showed that the patients at stage II-III progressed to BM more rapidly compared with those at stage I. It was worth nothing that an intensive postoperative BS particularly improved survival of patients at stage II. Consequently, it was rational to suggest patients with a heavy local tumor burden receive intensive postoperative BS screening. From an intrinsic biological point of view, early BC presents special metastatic behaviors [32,33], so postoperative monitoring strategies should vary accordingly. The ER-Her2-subtype, with a dramatically increased risk of distant relapse [34], accounted for 23.9% (253/1059) of patients in our study. An intensive postoperative BS improved OS as well as OSABM among ER-Her2-patients. Thus, we assumed that an intensive postoperative BS for ER-Her2-patients might be of significance. However, an intensive postoperative BS did not convert into a survival benefit in Her2+ patients. It is possible that this was due to limited Her2 status detection techniques and therapeutic options, even though early postoperative detection of BM was performed. In our study, 367 out of 1059 patients were diagnosed with BM before 2009, when Her2 status detection techniques were not commonly used in China, and trastuzumab was not widely implemented for relapse patients.

It is also worth noting that for all eligible patients, 26.5% (281/1059) were diagnosed with BM only, 37.6% (398/1059) were BM with VM, 23.6% (250/1059) were BM followed by VM, and 12.3% (130/1059) were VM followed by BM. There is probably a certain percent of patients classified as BM with VM who developed BM first and then progressed to VM but were not detected when simple BM originated. Previous studies showed that 26% to 50% of patients with early BC developed bone metastasis as the first site of distant relapse [4]. Consequently, early detection and treatment of BM may prolong the interval to visceral metastasis. As predicted, according to interaction and univariate stratified subgroup analysis, an intensive postoperative BS could improve OS for patients with ‘’BM to VM”, thus supporting the idea that early detection and early treatment are effective.

This multicenter real-world study showed that an intensive postoperative BS should probably be recommended as a follow-up strategy for patients with BC with BM. The main limitation of the present study is the retrospective study design. When evaluating the prognostic value of an intensive postoperative BS, cost-effectiveness and quality of life were not included in the analyses. Future studies with a randomized design are warranted to get an explicit estimation.

## 5. Conclusions

This multicenter real-world study showed that intensive postoperative BS screening improved survival for BC patients with BM, and should be recommended for postoperative surveillance, especially for patients at clinical high-risk.

## Figures and Tables

**Figure 1 cancers-14-05835-f001:**
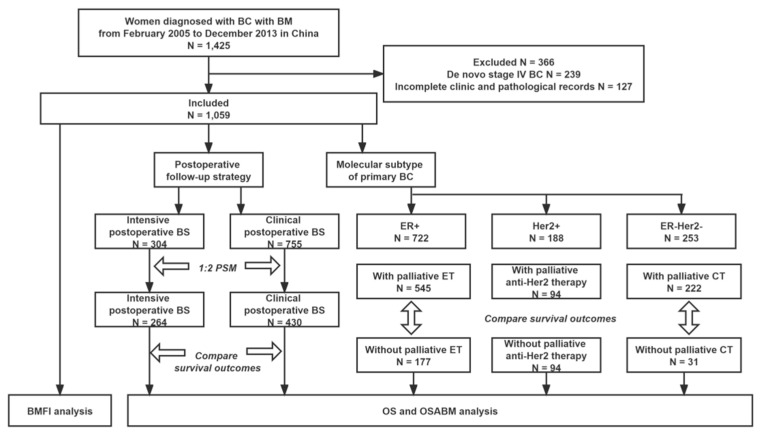
Flowchart of the process of patient’s enrollment and analyses. Abbreviations: BC = breast cancer; BM = bone metastasis; BMFI = bone metastasis-free interval; BS = bone scan; CT = chemotherapy; ER = estrogen receptor; Her2 = Human epidermal growth factor receptor 2; ET = endocrine therapy; OS = overall survival; OSABM = overall survival after diagnosis of bone metastasis; PSM = propensity scores matching.

**Figure 2 cancers-14-05835-f002:**
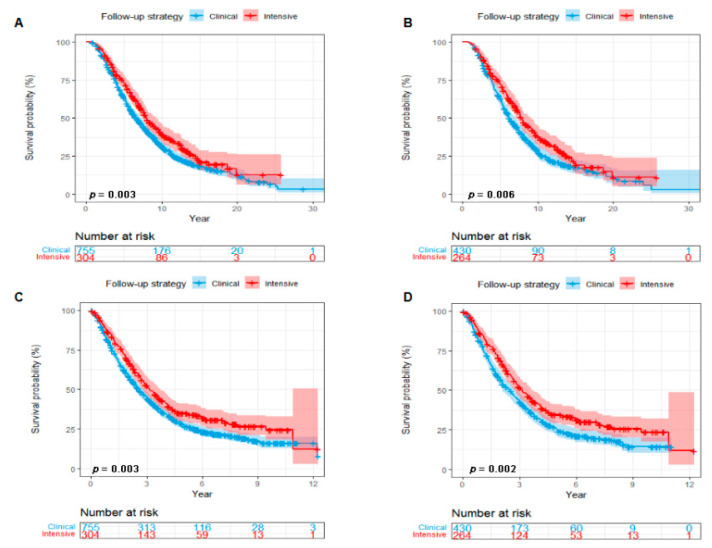
Kaplan–Meier curves showing a comparison of survival among patients with breast cancer with BM according to postoperative follow-up strategy (Intensive postoperative BS vs. Clinical postoperative BS). OS curves before (**A**) and after (**B**) PSM. OSABM curves before (**C**) and after (**D**) PSM. Abbreviations: BS = bone scan; BM = bone metastasis; OS = overall survival; OSABM = overall survival after diagnosis of bone metastasis; PSM = propensity scores matching.

**Figure 3 cancers-14-05835-f003:**
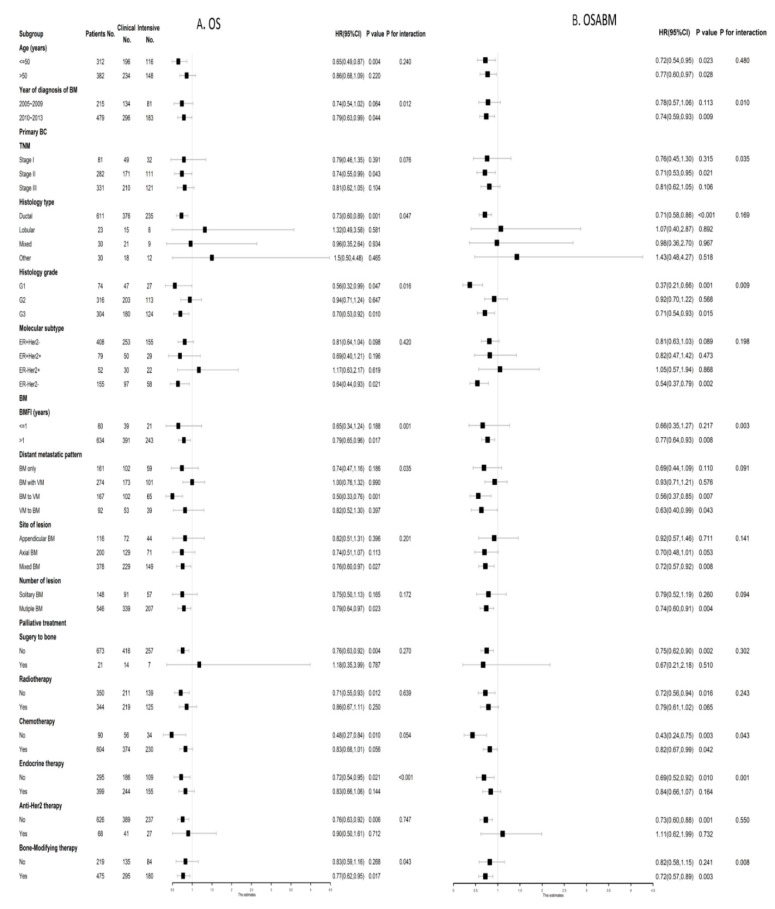
Forest plots of interaction and univariate subgroup analyses on the association between postoperative follow-up strategies (Intensive postoperative BS vs. Clinical postoperative BS) and (**A**) OS and (**B**) OSABM of patients with breast cancer with BM after PSM. Abbreviations: BS = bone scan; BM = bone metastasis; BMFI = bone metastasis-free interval; 95%CI = 95% confidence interval; ER = estrogen receptor; Her2 = Human epidermal growth factor receptor 2; HR = hazard risk; No. = Numbers of patients; OS = overall survival; OSABM = overall survival after bone metastasis; PSM = propensity scores matching.

**Figure 4 cancers-14-05835-f004:**
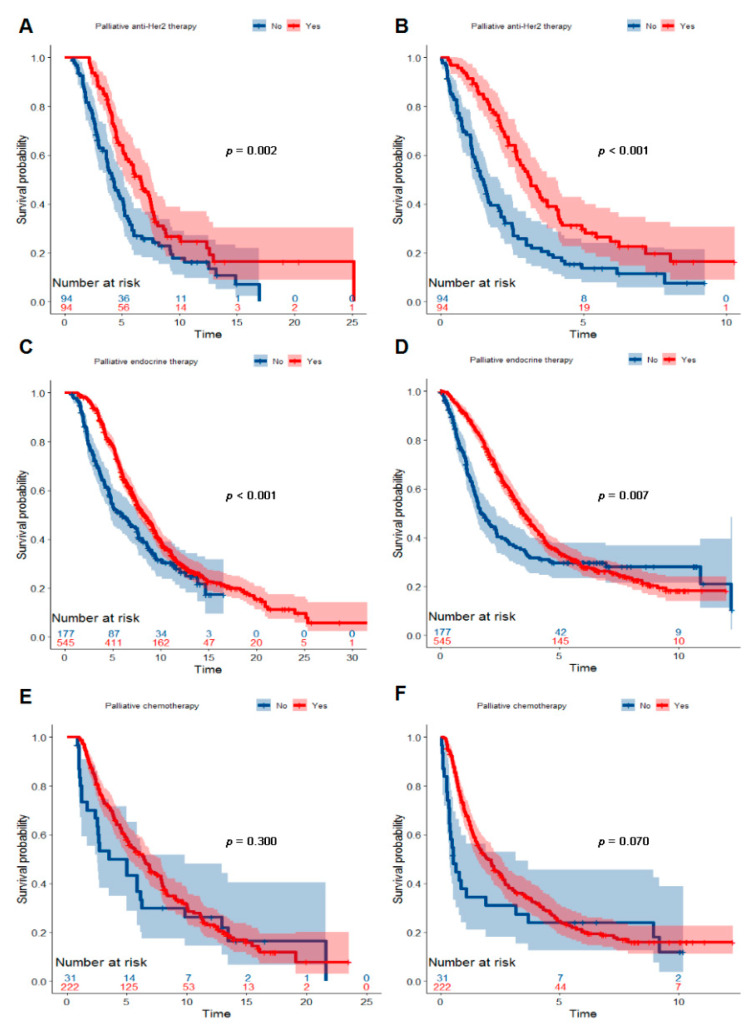
Kaplan–Meier curves showing a comparison of survival time among patients with breast cancer with BM according to molecular subtype and palliative treatment. Curves for OS (**A**) and OSABM (**B**) of patients with a Her2+ breast cancer stratified by palliative anti-Her2 therapy. Curves for OS (**C**) and OSABM (**D**) of patients with an ER+ breast cancer stratified by palliative endocrine therapy. Curves for OS € (**E**) and OSABM (**F**) of patients with an ER-Her2-breast cancer stratified by palliative chemotherapy. Abbreviations: BM = bone metastasis; ER = estrogen receptor; Her2 = Human epidermal growth factor receptor 2; OS = overall survival; OSABM = overall survival after diagnosis of bone metastasis.

**Figure 5 cancers-14-05835-f005:**
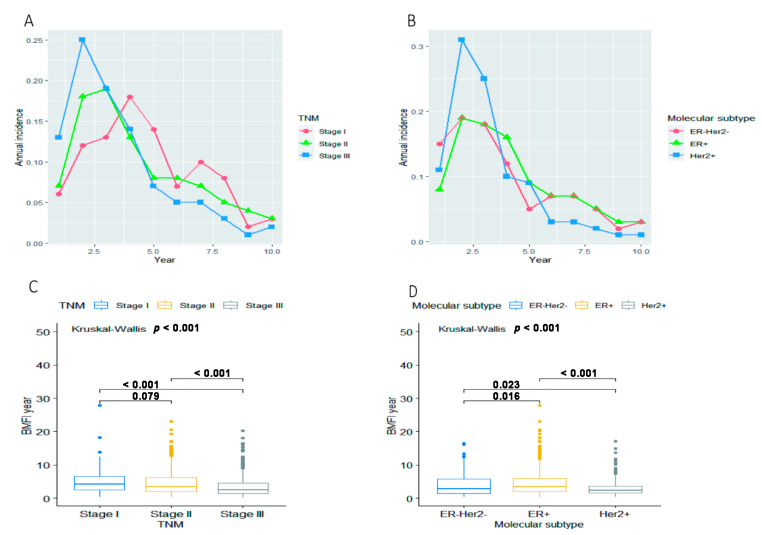
Annual incidence of BM for overall eligible patients (N = 1059) in groups stratified by (**A**) TNM stage and (**B**) molecular subtype. The distribution of BMFI for overall eligible patients (N = 1059) in groups stratified by (**C**) TNM stage and (**D**) molecular subtype. Abbreviations: BM = bone metastasis; BMFI = bone metastasis-free survival; ER = estrogen receptor; Her2 = Human epidermal growth factor receptor 2.

**Table 1 cancers-14-05835-t001:** Clinicopathological characteristics of eligible patients (N = 1059) stratified by postoperative follow-up strategy (Clinical postoperative BS vs. Intensive postoperative BS) before and after PSM.

Clinicopathological Characteristics		Before PSM				After PSM			
		Clinical *n* = 755	Intensive *n* = 304	P	SMD	Clinical *n* = 430	Intensive *n* = 264	P	SMD
Age * (mean (SD))		51.66 (10.42)	51.76 (11.04)	0.891	0.009	51.08 (10.32)	51.48 (11.06)	0.633	0.037
Year of diagnosis of BM (%)	2010~2013	474 (62.8)	218 (71.7)	0.007	0.191	296 (68.8)	183 (69.3)	0.961	0.010
	2005~2009	281 (37.2)	86 (28.3)			134 (31.2)	81 (30.7)		
Histology type (%)	Ductal	661 (87.5)	258 (84.9)	0.260	0.128	376 (87.4)	235 (89.0)	0.796	0.080
	Lobular	21 (2.8)	16 (5.3)			15 (3.5)	8 (3.0)		
	Mixed	31 (4.1)	13 (4.3)			21 (4.9)	9 (3.4)		
	Other **	42 (5.6)	17 (5.6)			18 (4.2)	12 (4.5)		
Histology grade (%)	G1	129 (17.1)	28 (9.2)	<0.001	0.406	47 (10.9)	27 (10.2)	0.417	0.103
	G2	374 (49.5)	117 (38.5)			203 (47.2)	113 (42.8)		
	G3	252 (33.4)	159 (52.3)			180 (41.9)	124 (47.0)		
TNM (%)	Stage I	89 (11.8)	38 (12.5)	0.886	0.033	49 (11.4)	32 (12.1)	0.744	0.060
	Stage II	314 (41.6)	129 (42.4)			171 (39.8)	111 (42.0)		
	Stage III	352 (46.6)	137 (45.1)			210 (48.8)	121 (45.8)		
Molecular subtype (%)	ER+Her2-	443 (58.7)	175 (57.6)	0.139	0.153	253 (58.8)	155 (58.7)	0.922	0.054
	ER+Her2+	74 (9.8)	30 (9.9)			50 (11.6)	29 (11.0)		
	ER-Her2+	51 (6.8)	33 (10.9)			30 (7.0)	22 (8.3)		
	ER-Her2-	187 (24.8)	66 (21.7)			97 (22.6)	58 (22.0)		
Distant metastatic pattern (%)	BM only	217 (28.7)	64 (21.1)	0.003	0.251	102(23.7)	59 (22.3)	0.780	0.081
	BM to VM	183 (24.2)	67 (22.0)			102 (23.7)	65 (24.6)		
	BM with VM	277 (36.7)	121 (39.8)			173 (40.2)	101 (38.3)		
	VM to BM	78 (10.3)	52 (17.1)			53 (12.3)	39 (14.8)		
BMFI (%)	≤1 year	82 (10.9)	21 (6.9)	0.064	0.139	39 (9.1)	21 (8.0)	0.713	0.040
	>1 year	673 (89.1)	283 (93.1)			391 (90.9)	243 (92.0)		
Site of osseous lesion (%)	Appendicular	122 (16.2)	55 (18.1)	0.204	0.123	72 (16.7)	44 (16.7)	0.653	0.072
	Axial	235 (31.1)	78 (25.7)			129 (30.0)	71 (26.9)		
	Mixed	398 (52.7)	171 (56.2)			229 (53.3)	149 (56.4)		
Number of osseous lesion (%)	Multiple	595 (78.8)	237 (78.0)	0.825	0.021	339 (78.8)	207 (78.4)	0.970	0.010
	Solitary	160 (21.2)	67 (22.0)			91 (21.2)	57 (21.6)		
Palliative treatment on BM									
Surgery to bone (%)	No	731 (96.8)	297 (97.7)	0.573	0.054	416 (96.7)	257 (97.3)	0.824	0.036
	Yes	24 (3.2)	7 (2.3)			14 (3.3)	7 (2.7)		
Radiotherapy (%)	No	357 (47.3)	168 (55.3)	0.023	0.160	211 (49.1)	139 (52.7)	0.402	0.072
	Yes	398 (52.7)	136 (44.7)			219 (50.9)	125 (47.3)		
Endocrine therapy (%)	No	333 (44.1)	128 (42.1)	0.599	0.040	186 (43.3)	109 (41.3)	0.667	0.040
	Yes	422 (55.9)	176 (57.9)			244 (56.7)	155 (58.7)		
Chemotherapy (%)	No	119 (15.8)	36 (11.8)	0.124	0.114	56 (13.0)	34 (12.9)	1.000	0.004
	Yes	636 (84.2)	268 (88.2)			374 (87.0)	230 (87.1)		
Anti-Her2 therapy (%)	No	698 (92.5)	267 (87.8)	0.023	0.155	389 (90.5)	237 (89.8)	0.868	0.023
	Yes	57 (7.5)	37 (12.2)			41 (9.5)	27 (10.2)		
Bone-Modifying therapy (%)	No	263 (34.8)	96 (31.6)	0.347	0.069	135 (31.4)	84 (31.8)	0.974	0.009
	Yes	492 (65.2)	208 (68.4)			295 (68.6)	180 (68.2)		

* Age at diagnosis of breast cancer with bone metastasis. ** Other histological types of invasive breast cancer in addition to infiltrating ductal or lobular carcinoma according to WHO criteria. Abbreviations: BM = bone metastasis; BMFI = bone metastasis-free interval; BS = bone scan; ER = estrogen receptor; Her2 = Human epidermal growth factor receptor 2; PSM = propensity scores matching; SD = standard deviation; SMD = standardized mean differences; VM = visceral metastasis.

**Table 2 cancers-14-05835-t002:** Univariate and multivariate analysis of clinicopathological factors affecting OS and OSABM among eligible patients (N = 694) after PSM.

Clinicopathological Factor	No.	Events	OS						OSABM					
			Univariate			Multivariate			Univariate			Multivariate		
			Crude HR	95%CI	Crude *p* Value	Adjusted HR	95%CI	Adjusted *p* Value	Crude HR	95%CI	Crude *p* Value	Adjusted HR	95%CI	Adjusted *p* Value
Follow-up strategy														
Clinical postoperative BS	430	326	0.77	0.64,0.93	0.006	0.77	0.64,0.93	0.006	0.75	0.63,0.90	0.002	0.71	0.60,0.86	<0.001
Intensive postoperative BS	264	175												
Age * (year)														
<=50	312	210	0.98	0.82,1.18	0.846	Not selected			1.26	1.06,1.51	0.011	1.23	1.03,1.47	0.026
>50	382	291												
Year of diagnosis of BM														
2005~2009	215	172	0.85	0.71,1.02	0.084	0.85	0.74,1.03	0.098	0.96	0.79,1.15	0.635	Not selected		
2010~2013	479	329												
Histology type					0.001			<0.001			0.025			0.002
Ductal	611	449	Ref.			Ref.			Ref.			Ref.		
Lobular	23	18	1.21	0.76,1.94	0.424	1.01	0.62,1.65	0.969	0.92	0.58,1.48	0.738	0.87	0.53,1.41	0.564
Mixed	30	20	0.85	0.54,1.34	0.487	0.76	0.48,1.21	0.254	0.94	0.60,1.47	0.787	0.79	0.50,1.26	0.325
Other **	30	14	0.32	0.18,0.55	<0.001	0.30	0.17,0.52	<0.001	0.44	0.26,0.74	0.002	0.36	0.21,0.62	<0.001
Histology grade					0.411						0.659			
G1	74	61	Ref.						Ref.					
G2	316	225	1.04	0.78,1.38	0.809	Not selected			0.92	0.70,1.23	0.583	Not selected		
G3	304	215	1.16	0.87,1.54	0.317				1.00	0.76,1.33	0.981			
TNM stage					<0.001			0.002			0.003			0.002
Stage I	81	58	Ref.			Ref.			Ref.			Ref.		
Stage II	282	192	1.09	0.81,1.46	0.584	1.09	0.81,1.46	0.593	1.09	0.81,1.46	0.582	1.19	0.89,1.60	0.247
Stage III	331	251	1.56	1.17,2.08	0.003	1.48	1.11,1.99	0.009	1.44	1.08,1.92	0.012	1.56	1.16,2.09	0.003
Molecular subtype					<0.001			0.178			<0.001			0.310
ER+Her2-	408	275	Ref.			Ref.			Ref.			Ref.		
ER+Her2+	79	60	1.47	1.11,1.95	0.007	1.23	0.92,1.64	0.156	1.23	0.93,1.63	0.145	0.95	0.72,1.27	0.739
ER-Her2+	52	42	2.53	1.82,3.51	<0.001	1.42	0.98,2.05	0.065	1.95	1.41,2.71	<0.001	1.27	0.89,1.83	0.194
ER-Her2-	155	124	1.50	1.21,1.86	<0.001	1.17	0.91,1.49	0.218	1.63	1.31,2.01	<0.001	1.21	0.95,1.55	0.126
Distant metastatic pattern					<0.001			<0.001			<0.001			<0.001
BM only	161	84	Ref.			Ref.			Ref.			Ref.		
BM with VM	274	227	2.04	1.59,2.63	<0.001	2.01	1.55,2.62	<0.001	2.36	1.83,3.04	<0.001	2.32	1.79,3.01	<0.001
BM to VM	167	110	1.34	1.01,1.79	0.042	1.34	1.00,1.81	0.052	1.22	0.91,1.62	0.181	1.27	0.95,1.71	0.108
VM to BM	92	80	1.64	1.20,2.23	0.002	1.89	1.37,2.61	<0.001	3.03	2.23,4.13	<0.001	3.43	2.49,4.73	<0.001
BMFI (year)														
≤1	60	45	0.29	0.21,0.39	<0.001	0.29	0.21,0.41	<0.001	0.75	0.55,1.02	0.062	0.80	0.58,1.10	0.799
>1	634	456												
Site of osseous lesion					0.014			0.090			0.001			0.078
Appendicular	116	75	Ref.			Ref.			Ref.			Ref.		
Axial	200	135	1.26	0.95,1.68	0.107	1.14	0.85,1.52	0.377	1.14	0.86,1.51	0.370	1.09	0.82,1.45	0.561
Mixed	378	291	1.45	1.12,1.87	0.004	1.38	1.02,1.85	0.035	1.52	1.18,1.96	0.001	1.36	1.01,1.81	0.040
Number of osseous lesion														
Solitary	148	101	1.29	1.04,1.61	0.022	0.98	0.74,1.30	0.908	1.37	1.10,1.71	0.005	1.12	0.85,1.48	0.437
Multiple	546	400												
Surgery to bone														
No	673	487	0.60	0.35,1.02	0.059	0.64	0.37,1.10	0.107	0.71	0.42,1.21	0.213	Not selected		
Yes	21	14												
Palliative radiotherapy														
No	350	242	1.10	0.92,1.31	0.289	Not selected			1.08	0.91,1.29	0.388	Not selected		
Yes	344	259												
Palliative endocrine therapy														
No	295	218	0.61	0.51,0.73	<0.001	0.62	0.50,0.78	<0.001	0.62	0.52,0.75	<0.001	0.68	0.55,0.85	0.001
Yes	399	283												
Palliative chemotherapy														
No	90	62	0.64	0.72,1.22	0.635	Not selected			0.94	0.72,1.23	0.668	Not selected		
Yes	604	439												
Palliative anti-Her2 therapy														
No	626	452	1.14	0.85,1.54	0.375	Not selected			0.85	0.63,1.14	0.273	Not selected		
Yes	68	49												
Bone-Modifying therapy														
No	219	141	1.04	0.86,1.26	0.697	Not selected			0.99	0.81,1.20	0.901	Not selected		
Yes	475	360												

* Age at diagnosis of breast cancer with bone metastasis. ** Other histological types of invasive breast cancer in addition to infiltrating ductal or lobular carcinoma according to WHO criteria. Abbreviations: BM = bone metastasis; BMFI = bone metastasis-free interval; BS = bone scan; 95%CI = 95% confidence interval; ER = estrogen receptor; Her2 = Human epidermal growth factor receptor 2; HR = hazard ratio; OS = overall survival; OSABM = overall survival after bone metastasis; PSM = propensity scores matching; Ref. = reference; VM = visceral metastasis.

## Data Availability

All data generated or analyzed during the study are included in the published paper.

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
