# Peer review of "Survival in Breast Cancer Patients with Bone Metastasis: A Multicenter Real-World Study on the Prognostic Impact of Intensive Postoperative Bone Scan after Initial Diagnosis of Breast Cancer (CSBrS-023)"

_cancers, 2022, doi:10.3390/cancers14235835_

Round 1

Reviewer 1 Report

This is an interesting article comparing the usefulness of bone scintigraphy as a routine postoperative surveillance for breast cancer patients. The study provides useful information, with data analyzed in detail.

The biggest concern is the patients included in this study. When bone scintigraphy is performed as a routine postoperative surveillance for breast cancer patients, the target population will be patients who may or may not subsequently develop bone metastases. Since the authors are examining the utility of an intervention in this patient population, I strongly believe that this study should not have been limited to patients with bone metastases, but should have included patients without bone metastases as well.

Author Response

Point 1: The biggest concern is the patients included in this study. When bone scintigraphy is performed as a routine postoperative surveillance for breast cancer patients, the target population will be patients who may or may not subsequently develop bone metastases. Since the authors are examining the utility of an intervention in this patient population, I strongly believe that this study should not have been limited to patients with bone metastases, but should have included patients without bone metastases as well.

Response 1: 

Thanks very much for taking the time to review this manuscript. We appreciate your comments and suggestions.

We can’t agree more with your suggestion about the best target population, which would be a huge cohort population, stratified by postoperative BS, including patients with and without bone metastasis. According to the data of the Danish population-based cohort study [1] and the SEER-medicare database [2], 4%-5.8% of breast cancer patients will develop BM. As this rate, we should probably include 25,000-35,000 patients for this analysis which is not available now. Your suggestions are constructive and provide the direction for our next research.

In our study, we retrospectively identified 1,425 patients with BM, focused on the survival difference by intensive BS or not. It’s not perfect but does show some clues. According to your suggestion , we extracted 231 patients without BM from our database and matched 907 patients with BM to show the sensitivity and stability of the findings in our study. We reached a conclusion consistent with the previous analysis (shown in Figure 1 and Figure 2).

Based on the current study design, we can't draw a firm conclusion that intensive postoperative BS screening improved survival for the whole BC patients but we do prove the survival benefit for BC patients with BM to some extent.

Reference

1.Cetin K, Christiansen CF, Sværke C et al. Survival in patients with breast cancer with bone metastasis: a Danish population-based cohort study on the prognostic impact of initial stage of disease at breast cancer diagnosis and length of the bone metastasis-free interval. BMJ Open 2015; 5(4): e007702. doi:10.1136/bmjopen-2015-007702.

2.Keating NL, Landrum MB, Guadagnoli E et al. Surveillance testing among survivors of early-stage breast cancer. J Clin Oncol 2007; 25(9):1074-1081. doi:10.1200/JCO.2006.08.6876.

Figures

Figure 1. Kaplan-Meier survival curves for OS of clinical high-risk breast cancer patients according to postoperative bone metastasis screening strategy (intensive vs. clinical) before PSM.

Figure 2. Kaplan-Meier survival curves for OS of clinical high-risk breast cancer patients according to postoperative bone metastasis screening strategy (intensive vs. clinical) after PSM.

Reviewer 2 Report

This paper demonstrated that intensive postoperative bone scan (BS) screening improved survival for BC patients with bone mets (BM) from retrospectively collected real-world data of five Chinese institutions.

The logic itself is consistent with the idea that an intensive BS can detect the development of BM earlier in high-risk patients and treat them earlier, thereby improving prognosis. In fact, retrospective analyses of a reasonable amount of data can produce solid results.

At any rate, as the authors described, it is a typical retrospective observational study, and as such, the contents forced me to make both plausible directions and vague analogies.

For example, palliative anti-Her2 therapy improved OS and OSABM of patients with a Her2+ BC. And authors speculated ‘this evidence suggested that early detection and effective early intervention for BM could be translated into survival benefit,’ however I do not help but feel uncomfortable with the simplistic conjecture proposed.

As stated at the beginning of the discussion, from the standpoint of molecular subtypes, ‘palliative systemic therapy improved both OS and OSABM among patients with a Her2+ and ER+ BC, indicating that improvements in treatments in the modern sophisticated medicine era could result in earlier detection and treatments of BM, thus promoting survival’ is far overstating the case.

The results of the detailed analyses (systemic treatment had prognostic-prolonging effects) do not provide direct evidence that the subject intensive BS contributed to its prognostic improvement. In this most critical part of the article, authors must be meticulous in stating their reasoning based on scientific evidence.

Although the limitations and biases of the data source itself cannot be excluded due to the observational database study, the richness of the number of original data from which the conclusions were drawn may be commendable.

Anyway, I hope authors can dispel my doubts, because the presentation of the paper itself could be very different without going through the important issues mentioned above.

Minor comment:

1.      In Figure 4 E&F, although the log-rank tests showed no significant difference, it should be concluded that short-term prognosis is improved by chemotherapy.

2.      In line 99, ‘was’ are duplicated.

3.      in line 311, ‘nothing’ is mistake for ‘noting’.

Author Response

Point 1:

At any rate, as the authors described, it is a typical retrospective observational study, and as such, the contents forced me to make both plausible directions and vague analogies.

For example, palliative anti-Her2 therapy improved OS and OSABM of patients with a Her2+ BC. And authors speculated ‘’this evidence suggested that early detection and effective early intervention for BM could be translated into survival benefit,’’ however I do not help but feel uncomfortable with the simplistic conjecture proposed.

As stated at the beginning of the discussion, from the standpoint of molecular subtypes, ‘’palliative systemic therapy improved both OS and OSABM among patients with a Her2+ and ER+ BC, indicating that improvements in treatments in the modern sophisticated medicine era could result in earlier detection and treatments of BM, thus promoting survival’’ is far overstating the case.

Response 1:

Thanks very much for taking the time to review this manuscript. We appreciate your comments and suggestions. We do have some over simplistic interpretation of these results. We have revised inappropriate expressions in the discussion section (Line 275-277).

As your comments, the logic is consistent with the idea that the intensive BS can detect BM earlier and promote early access to effective palliative systemic therapies, thereby improving survival outcomes of patients with BM.

As shown in Figure 4 in the manuscript, palliative endocrine and anti-Her2 therapy did improve the survival of patients with an ER-positive and Her2-positive phenotype, respectively. These results indicated that phenotype-specific palliative therapy was effective to improve the prognosis of patients with BM. From the forest plots of interaction and univariate subgroup analyses (shown in Figure 3 in the manuscript), the favorable prognostic value of intensive postoperative BS was significant in patients who underwent bone-modifying therapy but not in patients who did not. Notably, the long-term survival results of the ABCSG-18 trial were announced at the 2022 ASCO Congress, and dezumumab can significantly improve the survival of postmenopausal patients with hormone receptor-positive early BC [1]. For patients receiving palliative chemotherapy , anti-Her2 therapy, and endocrine therapy, the intensive BS showed a trend in survival benefits, but there was no statistical difference.

To better interpret the relationship between postoperative follow-up strategies, the survival, and the phenotype-specific palliative systemic treatments for BM patients with distinct molecular subtypes, we have further performed the interaction and univariate subgroup analyses (the results are shown in Figure 1 in the response letter). Different from the overall patients, the intensive BS improved the survival in ER-Her2- BC patients who received palliative chemotherapy. We know that this analysis may still not be perfect, and we will continue to explore this issue in future research.

Figure 1. Forest plots of interaction and univariate subgroup analyses on the association between postoperative follow-up strategies and survival of patients with BC with BM after PSM. Only the patients with an ER+ breast cancer were included in the palliative endocrine therapy subgroup. Only the patients with a Her2+ breast cancer were included in the palliative anti-Her2 therapy subgroup. Only the patients with an ER-Her2- breast cancer were included in the palliative chemotherapy subgroup.

Reference

  1. Gnant M, Pfeiler G, Steger GG,et al. Adjuvant denosumab in postmenopausal patients with hormone receptor-positive breast cancer (ABCSG-18): disease-free survival results from a randomised, double-blind, placebo-controlled, phase 3 trial. Lancet Oncol. 2019 Mar;20(3):339-351. doi: 10.1016/S1470-2045(18)30862-3. Epub 2019 Feb 19.

Point 2:

The results of the detailed analyses (systemic treatment had prognostic-prolonging effects) do not provide direct evidence that the subject intensive BS contributed to its prognostic improvement. In this most critical part of the article, authors must be meticulous in stating their reasoning based on scientific evidence.

Response 2:

We appreciate your comments and suggestions. As stated in Response 1, we have tried to prove the conjecture as far as possible that “intensive postoperative BS promotes early effective palliative therapies, thereby improving survival” based on current data by interaction and univariate subgroup analyses. Even though imperfect, at least it provides some clues. Thank you again for putting forward the direction of our next research.

Point 3: 

In Figure 4 E&F, although the log-rank tests showed no significant difference, it should be concluded that short-term prognosis is improved by chemotherapy.

Response 3:

Thanks for your suggestion. In Figure 4 E&F, the survival curves of OS and OSABM crossed at about 10 and 5 years, respectively, which showed that the palliative chemotherapy did not convert into long-term survival benefits in ER-Her2-patients. According to your comments, we calculated the 5-year OS rate and the 2-year OSABM rate in the two groups of patients, found that palliative chemotherapy increased 5-year OS by 14.3% (57.7% vs 43.4%) and 2-year OSABM by 18.7% (49.7% vs 31.0%) compared with patients without palliative chemotherapy. We have added the detailed data and taken brief comments to the discussion section of the manuscript (Line 312-314).

Point 4: 

In line 99, ‘was’ are duplicated.

Response 4:

We have revised it in Line 100: was was.

Point 5: 

In line 311, ‘nothing’ is mistake for ‘noting’.

Response 5:

We have revised it in Line 335: noting to nothing.

Round 2

Reviewer 1 Report

The authors have revised their original manuscript partly according to the reviewers’ comments. In some points, the authors decided to keep their contents unchanged, however, their rebuttal seems almost reasonable. I would think that this revised manuscript is better organized and suitable for publication.

Author Response

Thanks very much for taking the time to review this manuscript and your recognition of our work.

Reviewer 2 Report

The authors inevitably focused on the prognostic benefits of salvage systemic therapy, and unfortunately were unable to show evidence that the implementation of the crucial intensive bone scintigraphy that I pointed out was associated with improved prognosis.

Author Response

Thanks very much for taking the time to review this manuscript. Our study aimed to explore the prognostic factors of patients with bone metastases, including intensive BS and the palliative treatment strategies. The results showed that early asymptomatic bone scan screening (intensive BS) and effective palliative treatments improved the long-term prognosis of breast cancer patients with bone metastases. Your concern was whether intensive BS could improve survival outcomes by promoting early treatment, so we performed an exploratory subgroup analysis and a supplementary note. The favorable prognostic value of intensive BS was significant in patients who underwent bone-modifying therapy but not in patients who did not (detailed in Figure 3 in the manuscript). Moreover, the intensive BS improved the survival in ER-Her2- BC patients who received palliative chemotherapy but not in patients who did not (shown in Figure 1 in the response letter). The intensive BS itself is an early screening of asymptomatic stage. The median time when patient received the first intensive BS was 2.5 years after initial diagnosis of BC. However, guidelines recommend BS for 5 years or for bone-related symptoms. Based on these evidence, we have proved your concern: early and asymptomatic BS (intensive BS) could promote effective treatment, thus improved survival. Thanks again for your advice.